Allele specific expression in worker reproduction genes in the bumblebee Bombus terrestris

Amarasinghe Harindra E. 1
Toghill Bradley J. 2
Nathanael Despina 1
Mallon Eamonn B. 1 ebm3@le.ac.uk
1 Department of Biology, University of Leicester , UK
2 Department of Cardiovascular Sciences, University of Leicester , UK
Higley Leon
Electronic publication date: 2015 Jul 14
Publication date: 2015
Volume: 3
Electronic Location ID: e1079
Received 2015 Mar 8; Accepted 2015 Jun 14
Copyright: © 2015 Amarasinghe et al.
Copyright year: 2015
Copyright holder: Amarasinghe et al.
License: This is an open access article distributed under the terms of the Creative Commons Attribution License, which permits unrestricted use, distribution, reproduction and adaptation in any medium and for any purpose provided that it is properly attributed. For attribution, the original author(s), title, publication source (PeerJ) and either DOI or URL of the article must be cited.
License URL: https://creativecommons.org/licenses/by/4.0/

Keywords: Social insect, Hymenoptera, Ecdysone, Epigenetics

Funding: NERC NE/H010408/1 This work was financially supported by NERC grant no. NE/H010408/1 to EBM. The funders had no role in study design, data collection and analysis, decision to publish, or preparation of the manuscript.

==============================
Methylation has previously been associated with allele specific expression in ants. Recently, we found methylation is important in worker reproduction in the bumblebee Bombus terrestris. Here we searched for allele specific expression in twelve genes associated with worker reproduction in bees. We found allele specific expression in Ecdysone 20 monooxygenase and IMP-L2-like. Although we were unable to confirm a genetic or epigenetic cause for this allele specific expression, the expression patterns of the two genes match those predicted for imprinted genes.

Introduction

Epigenetics refers to heritable changes in gene expression that do not involve DNA sequence alterations. Several recent reviews have heralded hymenopteran insects (ants, bees and wasps) as important emerging models for epigenetics (Glastad et al., 2011; Weiner & Toth, 2012; Welch & Lister, 2014; Yan et al., 2014). This is based mainly on data showing a fundamental role for methylation in their biology (Chittka, Wurm & Chittka, 2012). Methylation, the addition of a methyl group to a cytosine, is an epigenetic marker in many species (Glastad et al., 2011).

The recently sequenced genome of the bumblebee, Bombus terrestris displays a full complement of genes involved in the methylation system (Sadd et al., 2015). In a recent paper (Amarasinghe, Clayton & Mallon, 2014), we showed that methylation is important in worker reproduction in this bumblebee. We found methylation differences between the genomes of queenless reproductive workers and queenless non-reproductive workers. In a follow up experiment, queenless workers whose genomes had experimentally altered methylation (fed 5-aza-2′-deoxycytidine) were more aggressive and more likely to develop ovaries compared with control queenless workers.

Previous work has found methylation associated with allele specific expression in a number of loci in the ants Camponotus floridanus and Harpegnathos saltator (Bonasio et al., 2012). Based on our result showing the importance of methylation in bumblebee worker reproduction, we searched for allele specific expression in worker reproduction genes. We chose twelve genes previously associated with worker reproduction in bees (see Table 1). We looked for polymorphisms in their exonic DNA in queens and their daughter workers using single strand conformation polymorphism (SSCP) and Sanger sequencing.

SSCP relies on the principle that the electrophoretic mobility of a single-stranded DNA molecule is dependent on its structure (nucleotide sequence) and size. In the absence of the complementary strand, DNA becomes unstable and reanneals to itself to form conformations; hairpins, pseudoknots and triple helices (Nielsen, Novoradovsky & Goldman, 1995). These conformations vary according to the primary sequence of the molecule, such that a single nucleotide difference in DNA could dramatically affect the strand’s mobility through a gel due to its unique 3D structure.

If we found that the workers possessed an allele from the queen and another allele not present in the queen, this allele must be from the father. That is, we identified a matrigene (allele from the mother) and patrigene (allele from the father) at this locus. If we found this, we carried out an allele specific qPCR to ascertain if this locus displayed allele specific expression.

Methods

Identification of candidate genes and designing primers

Twelve social insect genes previously associated with differential expression in queens, reproducing workers and non-reproducing workers were selected via a literature search (Table 1). Sequences for all selected candidate genes were obtained from Apis mellifera genome data, available in NCBI. Apis mellifera data was BLASTed against the Bombus terrestris Nucleotide library (NCBI) in order to find the homolog in Bombus terrestris. Primers were designed to the exonic regions using Geneious Pro (version 5.5.6) and primer 3 version 0.4.0 (http://frodo.wi.mit.edu). The focus was on exonic regions to ensure that the same loci was present in the cDNA for the allele specific qPCR analysis.

Table 1 Candidate genes selected from the literature search.

Apis mellifera	Bombus terrestris	Biological function	
Chymotrypsin	Chymotrypsin-1-like (LOC100648122)	Upregulated in bumblebee non-reproductive workers (Pereboom et al., 2005)	
Gemini	Upstream binding protein 1-like (LOC100650338)	Upregulated in honeybee reproductive workers (Jarosch et al., 2011)	
Cabut	Zinc finger protein 691-like (LOC100642767)	Upregulated in honeybee non- reproductive workers (Cardoen et al., 2011)	
Ecdysone 20 monooxygenase	Ecdysone 20 monooxygenase-like (LOC100649449)	Upregulated in honeybee reproductive workers (Cardoen et al., 2011)	
Yolkless	Vitellogenin receptor-like (LOC100649042)	Upregulated in honeybee reproductive workers (Cardoen et al., 2011)	
Epidermal growth factor receptor	Epidermal growth factor receptor like (LOC100645521)	Upregulation of EGFR initiates ovary activation in queenless honeybee workers (Formesyn et al., 2014)	
Ribosomal Protein L26	Ribosomal Protein L26 like (LOC100648461)	Differentially expressed in honeybee reproductive and non-reproductive workers (Thompson et al., 2007)	
Odorant receptor2	Or2 odorant receptor 2/Queen mandibular pheromone (QMP) co-receptor (LOC100631089)	Upregulated in honeybee sterile workers (Grozinger et al., 2007)	
Dop3 D2-like dopamine receptor	D2 like dopamine receptor (LOC100644210)	Upregulated in honeybee non-reproductive workers (Vergoz, Lim & Oldroyd, 2012)	
Megator	Megator TPR like nucleoprotein (LOC100645723)	Upregulated in honeybee reproductive workers (Cardoen et al., 2011)	
Ecdysteroid regulated gene E93/Mblk-1 transcription factor	Mushroom body large-type Kenyon cell specific protein 1-like (LOC100645656)	Upregulated in honeybee reproductive workers (Cardoen et al., 2011)	
Ecdysone inducible gene L2/ImpL2	Neural/ectodermal development factor IMP-L2-like (LOC100645498)	Upregulated in honeybee non-reproductive workers (Cardoen et al., 2011)	
Notes.

NCBI gene IDs are in parentheses.

Samples

The queen and 5 randomly selected workers from each colony were used for SSCP analysis. Most candidate genes were tested in four different bumblebee colonies. Chymotrypsin, Gemini, Cabut and Yolkless were tested in colonies 1–4. Another four colonies (5–8) were used to test Epidermal growth factor receptor, Ribosomal protein L26, Odorant receptor 2, Dop3, Megator, Ecdysteroid regulated gene E93 (Mblk1) and Ecdysone inducible gene L2 (IMP-L2). Ecdysone 20 monooxygenase-like was tested in eight colonies. All qPCR data are based on bees from colony 5.

DNA and RNA extraction and cDNA synthesis

Bees were frozen in liquid nitrogen and then stored at −80 °C. Genomic DNA for SSCP analysis was extracted from each queen and respective worker bees using the Qiagen DNA Micro kit according to manufacturer’s instructions. Concentration of total genomic DNA was measured using the NanoDrop 1000 Spectrophotometer.

A 30 mg sample of frozen tissue was ground with mortar and pestle on dry ice. RNA was extracted with the QIAGEN RNeasy Mini Kit according to manufacturer’s instructions.

Any DNA contamination present in the above RNA extractions was removed according to Amplification Grade DNase I Kit protocol (Sigma-Aldrich), prior to the synthesis of cDNA. Concentrations of DNase treated RNA was determined by the NanoDrop 1000 spectrophotometer.

cDNA was synthesized from a 8 μl sample of RNA using the Tetro cDNA synthesis Kit (Bioline) as per manufacturer’s instructions. Synthesized cDNA was stored at −80 °C.

PCR amplifications

For each primer set, a 25 µl reaction volume (60 ng of DNA, 12.5 µl YB-Taq 2x Buffer, 1.5 µl of each forward and reverse primer (10 µlM/µl), 1 µl of 10 mM MgCl2 and 6.5 µl of dH20) was run using the following conditions: an initial denaturation for 5 min at 94 °C, 30 cycles of 30 s at 94 °C, 30 s each at the relevant annealing temperature followed by a final extension of 10 min at the relevant extension temperature and a holding step of 4 °C. The sequences and annealing and extension temperatures used for each primer set are in Table S1.

Prior to SSCP analysis, each PCR product (10 µl) was checked on a 3% agarose gel. If the correct size of amplicon was obtained, then the rest of the sample (15 µl) was used for SSCP.

SSCP analysis

SSCP analysis was carried out according to Gasser et al. (2007) using GMA wide mini S-2x25 gels (Elchrom Scientific). Sample denaturing solution was prepared by mixing 990 µl of 95% formamide with 10 µl of 1 M NaOH just prior to use. 4 µl of the PCR product was denatured with 7 µl of denaturing mixture, incubated in a thermocycler at 94 °C for 10 min and immediately chilled on ice for 5 min.

The temperature of the running buffer (1x TAE) in the Origins gel tank (Elchrom Scientific) was kept constant at 9 °C. 7 µl of the denatured PCR product was mixed with 2 µl of Elchrom loading dye and loaded in to a well on the gel. The gels were run at 72 V. The electrophoretic running times were varied depending on the fragment size; 10 h for 150–200bp fragment length, 12 h for 200–250bp fragment length, 15 h for 250–350bp fragment length and 17 h for 350–450bp fragment length.

Following electrophoresis, the gels were stained for 30 min with SybrGold (Invitrogen) (1:10,000 diluted in TAE) and destained with 100 ml of 1x TAE buffer for a further 30 min.

If a polymorphic banding pattern among the queen and her 5 workers was observed during SSCP, another SSCP was run to confirm the reproducibility of those results. The genomic DNA of those queen and worker bees were amplified with their respective primers (see Table S1) and PCR products were sent for commercial clean up and Sanger sequencing.

All sequencing results were blasted against NCBI, Bombus terrestris nucleotide library to verify if the correct sequence was amplified. Sequencing results were analyzed using the heterozygote analysis module in Geneious version 7.3.0 to identify heterozygotic nucleotide positions.

Allele specific PCR

Allele specific PCR was used to confirm the maternal and paternal alleles identified during heterozygote analysis. Allele specific primers were designed using Batch primer 3 program (http://probes.pw.usda.gov/batchprimer3/) to cover the heterozygotic nucleotide positions identified above. Two forward primers specific to either maternal (F1) or paternal (F2) allele sequences and a common reverse primer were designed. Genomic DNA of the queen and 5 heterozygous workers in each colony, were PCR amplified with these allele specific primers (Table 2). PCR products were checked on a 3% agarose gel. When using allele specific primers, only the allele which includes the relevant snp would be amplified. Primers which amplified the snp region successfully were used for qPCR analysis.

Allele specific quantitative PCR

Reference primers were designed according to Gineikiene, Stoskus & Griskevicius (2009). A common forward primer was designed to the same target heterozygote sequence, upstream of the heterozygote nucleotide position, leaving the same common reverse primer previously used with allele specific primers (see reference sequences in Table 2). The reference primers measure the total expression of the gene, whereas the allele specific primers measure the amount of expression due to the allele. Thus the ratio between the allele specific expression and reference locus expression would be the relative expression due to the allele.

Each heterozygous locus was run for 3 different reactions; maternal (F1), paternal (F2) and reference (Table 2). Three replicate samples were run for each reaction. All reactions were prepared by the Corbett robotics machine, in 96 well qPCR plates (Thermo Scientific, UK). The qPCR reaction mix (20 µl) was composed of 1 µl of diluted cDNA (50 ng/µl), 1 µl of forward and reverse primer (5 µM/µl each, Table 2), 10 µl 2x SYBR Green JumpStart Taq ReadyMix (Sigma Aldrich, UK) and 7 µl ddH20. Samples were run in a PTC-200 MJ thermocycler. The qPCR profile was; 4 min at 95 °C denaturation followed by 40 cycles of 30 s at 95 °C, 30 s at the relevant annealing temperature (Table 2) and 30 s at 72 °C and a final extension of 5 min at 72 °C.

Forward primers are different, both in their terminal base (to match the snp) and in their length. It is entirely possible that they may amplify more or less efficiently even if there was no difference in amount of template (Pfaffl, 2001). To test for this we repeated all qPCRs with genomic DNA 1 µl of diluted DNA (20 ng/ µl) from the same bees as the template. We would expect equal amounts of each allele in the genomic DNA. We also measured efficiency of each reaction as per Liu & Saint (2002).

Table 2 Allele specific primers used for gene expression analysis.

Gene	Heterozygote position	Primer sequence (5′-3′)	TA (°C)	Product size (bp)	
Ecdysone 20 monooxygenase like	48 (A/G)	F1: GCGGAAGCCGTCAGG	58	34	
F2: TTAGCGGAAGCCGTCAGA	
R: GCGAGGCCGTAAAGTGTAT	
Ecdysone 20 monooxygenase like internal reference		F: GATTTAGCGGAAGCCGTCAG	59	36	
	R: GCGAGGCCGTAAAGTGTAT	
IMP-L2-like	253 (A/G)	F1: ACTTGCCAAGCCAAGTCTG	59.5	205	
F2: CACTTGCCAAGCCAAGTCTA	
R: TTCGAGCCACTTCCTTTTCG	
IMP-L2-like internal reference		F: CTACACTTGCCAAGCCAAGTCT	59.5	207	
	R: TTCGAGCCACTTCCTTTTCG	
Notes.

The snp present is located at the ′ end of each forward primer (marked in red).

F1 Forward primer 1

F2 Forward primer 2

R Common reverse primer

TA Annealing temperature

Data analysis

Median Ct was calculated for each set of three technical replicates. A measure of relative expression (ratio) was calculated for each parental allele in each worker bee as follows: (1) ratiomaternal=Ematernal−CtmaternalEreference−Ctreference

(2) ratiopaternal=Epaternal−CtpaternalEreference−Ctreference.

E is the median efficiency of each primer set (Liu & Saint, 2002; Pfaffl, 2001). Matched paired t-tests was performed to check if the allele specific expression values are significantly different among the two parental alleles. All statistical analysis was carried out using R (3.1.0) (R Core Team , 2015).

Results

SSCP analysis

Exon coverage for each gene is given in Table 3. We found no polymorphisms in nine genes out of the twelve candidate genes tested (see Table 3). Figure 1 shows representative examples of these gels with the queen and her workers sharing the same banding pattern.

Figure 1 The queen (Q) and 5 workers (W1–W5) are represented in each colony.

SSCP gel results of six genes (A–F) with no queen-worker variations (homozygous banding patterns).

Table 3 SSCP exon coverage.

Percentage exon coverage for each gene was calculated as the sum of all tested amplicon lengths as a fraction of the total length of mRNA per gene.

Gene name	Coverage (%)	Polymorphism	
Chymotrypsin-1-like	92	Absent	
Upstream-binding protein 1-like	93	Absent	
Zinc finger protein 691-like	70	Absent	
Vitellogenin receptor-like	30	Absent	
Epidermal growth factor receptor like	21	Absent	
60S Ribosomal Protein L26 like	32	Absent	
Or2 odorant receptor 2	25	Absent	
D2 like dopamine receptor	54	Absent	
Megator TPR like nucleoprotein	17	Absent	
Ecdysone 20-monooxygenase-like	37	Present	
Mushroom body large-type Kenyon cell-specific protein 1-like	35	Present	
Neural/ectodermal development factor IMP-L2-like	47	Present	

Compared with the queen, workers in colony 3, 5 and 7 showed a heterozygous banding pattern in 3 genes; ecdysone 20-monooxygenase-like (ecdysone 20-monooxygenase-like), IMPL-2-like and MBLK1-like (Fig. 2).

Figure 2 SSCP allelic polymorphism in IMP-L2-like, MBLK1-like and Ecdysone 20-monooxygenase-like.

Sequences of polymorphic loci

We sequenced the three loci showing heterozygous banding patterns in SSCP gels. In ecdysone 20-monooxygenase-like, the queen sequence is homozygous (Fig. 2). At the snp (2,474th base pair of LOC100649449) the queen has a guanine (G), while all of her workers show double peaks corresponding to both guanine (G) and adenine (A) bases (see Table S2 for sequences). From this we identified the matrigene as containing guanine and the patrigene as containing adenine. A similar SSCP banding pattern was found for IMP-L2-like (Fig. 2). Again the queen had a guanine whereas workers contained a guanine and an adenine, this time at the 5,130th base pair of LOC100645498 (Table S2). MBLK1-like has several snps in the amplified region (Fig. 2 and Table S2).

Allele specific PCR

Allele specific primers designed for ecdysone 20-monooxygenase-like and IMP-L2-like worked successfully with genomic DNA and cDNA to produce the expected fragment lengths. They were used for gene expression analysis in the next section. Amplification of MBLK1-like using allele specific primer sets was unsuccessful possibly due to the large number of snps. Thus, we did not continue with qPCR analysis for MBLK1-like.

Allele specific qPCR

The allele specific primer sets for ecdysone 20-monooxygenase-like showed no difference in their ability to amplify genomic DNA (paired t-test: t = 0.4815, df = 4, p-value = 0.6553, Maternal primers Ct = 36.73 ± 2.494, Paternal primers Ct = 36.27 ± 1.792 (mean ± standard deviation)). For IMP-L2-like, the two primer sets did show a significant difference in efficiency to amplify genomic DNA (paired t-test: t = 7.062, df = 4, p-value = 0.002121, Maternal primers Ct = 35.83 ± 1.463, Paternal primers Ct = 32.49 ± 1.327 (mean ± standard deviation)). As Ct decreases with increasing copy number, the paternal primers amplified better than the maternal set.

To control for this difference in primer efficiency we used a modification of the pfaffl method to calculate a measure of expression (Liu & Saint, 2002; Pfaffl, 2001). This includes efficiency in its calculations (see ‘Methods’). The patrigene showed significantly increased expression compared to the matrigene in ecdysone 20-monooxygenase-like (Fig. 3) (paired t-test: t = − 7.517, df = 4, p-value = 0.001676). For IMP-L2-like, the matrigene was more expressed than the patrigene (Fig. 4) (paired t-test: t = 3.409, df = 4, p-value = 0.02705).

Figure 3 Expression (measured as ratio of allelic to reference primer expression (see ‘Methods’)) differences between maternal and paternal alleles of ecdysone 20-monooxygenase-like.

The first plot shows the expression data as individual dots. The diagonal lines join data from the same bee. Boxplots represent the distributions. The second plot shows the proportion of maternal to paternal expression as individual dots.

Figure 4 Expression (measured as ratio of allelic to reference primer expression (see ‘Methods’)) differences between maternal and paternal alleles of Ecdysone-inducible gene L2 (IMP-L2-like).

The first plot shows the expression data as individual dots. The diagonal lines join data from the same bee. Boxplots represent the distributions. The second plot shows the proportion of maternal to paternal expression as individual dots.

Discussion

Using a candidate gene approach we found evidence for allele specific expression in the bumblebee, Bombus terrestris. Out of twelve genes examined during this study, we found exonic variation in only three genes; MBLK1-like, IMP-L2-like and ecdysone 20-monooxygenase-like. Of these we were able to carry out allele specific qPCR on IMP-L2-like and ecdysone 20-monooxygenase-like. We found allele specific expression in ecdysone 20-monooxygenase-like and IMP-L2-like.

Use of SSCP to find exonic variation is challenging. SSCP detects variation in fragments up to 500bp size with a high resolution of 1bp. However, the sensitivity of SSCP decreases when the fragment length exceeds 200bp (Weber et al., 2005). Thus medium length fragments around 200bp were used for this analysis. Covering the full exome using SSCP would be a time consuming and labour intensive process. Added to this, variation in protein coding exons is expected to be rarer than in introns (Castle, 2011). One possibility would be to look at the exons in untranslated regions (UTRs), which would be expected to be more variable than protein coding exons (Araujo et al., 2012; Lytle, Yario & Steitz, 2007).

Our expression analysis used the bees’ whole bodies. Therefore gene expression patterns observed during this analysis should represent the overall expression of all body tissues. However, potentially it means allele specific expression which is only found in some tissues would be masked by the overall response.

We found allele specific expression in Ecdysone 20 monooxygenase. Ecdysone 20 monooxygenase catalyses the reaction which turns ecdysteroid ecdysone into 20-hydroxyecdysone, also an ecdysteroid. An up-regulation of ecdysone 20-monooxygenase-like was observed in egg laying honeybee workers compared to non-reproductive workers (Cardoen et al., 2011). Generally, ecdysteroids have been identified as key regulators of B. terrestris worker reproduction (Geva, Hartfelder & Bloch, 2005). Ecdysteriods are key compounds involved in ovary activation, regulating agonistic behaviour and establishing the dominance hierarchy in workers and queens (Geva, Hartfelder & Bloch, 2005). We also found allele specific expression in IMP-L2-like. In honeybees, this gene is linked with reproductive inhibition of workers. It functions similarly to an insulin like peptide and negatively regulates insulin signaling pathways to repress ovary activation (Cardoen et al., 2011; Grozinger et al., 2007).

Our analysis found allele specific expression in two worker reproduction genes. We are interested in this as an example of epigenetics. However, allele specific expression is known to be caused by a number of genetic (i.e., cis-acting inherited variation) as well as epigenetic (e.g., genomic imprinting) processes (Palacios et al., 2009). As our results are based on a single genetic line (colony) we are unable to say whether the examples of allele specific expression we found are due to epigenetic or genetic causes.

Given this, it is still interesting to note that the expression patterns of both ecdysone 20-monooxygenase-like and IMP-L2-like are consistent with those predicted for genomic imprinted genes involved in worker reproduction in a singly mated social insect colony (Queller, 2003). Queller (2003) used Haig’s kinship theory for the evolution of genomic imprinting (Haig, 2000) to predict the imprinting patterns of genes involved in various functions under various social contexts in the social insects. He predicted that genes that are associated with the initiation of worker reproduction (e.g., ecdysone 20-monooxygenase-like) should be paternally expressed in social insect species such as B. terrestris with singly-mated, monogynous (one queen), queenright (queen still alive) colonies. Ecdysone 20-monooxygenase-like’s expression is consistent with increased paternal expression. Reciprocally we would expect a gene that inhibits worker reproduction (e.g., IMP-L2-like) to be maternally expressed. IMP-L2-like’s expression is consistent with increased maternal expression. Fascinating as this is, it must be tempered with the proviso that, as previously stated, this work was carried out on a single genetic line so cannot differentiate epigenetic from genetic causes.

Clearly the candidate gene approach is limited in its application. Next generation sequencing technology allows gene expression analysis at genome-wide scale (RNA-seq). Several recent papers have applied RNA-seq to search, without bias, for novel imprinted genes in mammals (Okae et al., 2012; DeVeale, Van der Kooy & Babak, 2012; Gregg et al., 2010; Wang & Clark, 2014) and flowering plants (Gehring, 2013). Our results suggest that this could be a potentially fruitful avenue for research in social insects.

Supplemental Information

Table S1 Details of primers sets used in PCR reactions

Click here for additional data file.

Table S2 Sequences of loci for which there was heterozygote workers

Snps are in red.

Click here for additional data file.

Supplemental Information 3 qPCR dataset

Click here for additional data file.

Thanks to Sally Adams for discussions.

Additional Information and Declarations

Competing Interests

Author Contributions

The authors declare there are no competing interests.

Harindra E. Amarasinghe conceived and designed the experiments, performed the experiments, analyzed the data, wrote the paper, prepared figures and/or tables, reviewed drafts of the paper.

Bradley J. Toghill conceived and designed the experiments, performed the experiments, analyzed the data, reviewed drafts of the paper.

Despina Nathanael performed the experiments, reviewed drafts of the paper.

Eamonn B. Mallon conceived and designed the experiments, analyzed the data, wrote the paper, prepared figures and/or tables, reviewed drafts of the paper.

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
