# Peer review of "Allele specific expression in worker reproduction genes in the bumblebee Bombus terrestris"

_PeerJ, doi:10.7717/peerj.1079_

## Round 0.1 · original submission · Major Revisions

Drs Amarasinghe, Toghill, and Mallon, as you see from the accompanying reviews, your manuscript garnered diverse opinions regarding suitability. I find the comment of reviewers 2 and 3 most critical, and these offer specific points of concern regarding methods and interpretation. From my perspective, the key issues are justifying your methods and establishing whether or not they actually support genetic imprinting. Here, I am at the mercy of the reviewers, as this is not an area in which I have expertise. From their comments, I think additional justification in the manuscript and a response to specific points is warranted. In making the final decision with your revised manuscript, it is important that you address (either in the manuscript or your rebuttal) the points raised by each reviewer. I also this the manuscript would be strengthened by additional references as suggested by reviewer 2, particularly as regards Haig's theory.

Reviewer 1 ·

Basic reporting

None

Experimental design

None

Validity of the findings

My only comment for improvement is to emphasize for all readers the importance of this work. Is the last sentence helps some, saying this may be a worthy avenue for further investigation, but for readers like myself the data doesn’t tell me why. Please consider including some language linking the data and discussion of the data (maybe a sentence or two) for those not in the know of why this work is important. This will help all readers understand and emphasize the value of your work.

Comments for the author

This is a very well-written paper with a solid approach to the study. I am not a geneticist, but I was able to understand and enjoy the paper. This manuscript is worthy of publication and deserves publishing here in PeerJ. My only comment for improvement is to emphasize for all readers the importance of this work. Is the last sentence helps some, saying this may be a worthy avenue for further investigation, but for readers like myself the data doesn’t tell me why. Please consider including some language linking the data and discussion of the data (maybe a sentence or two) for those not in the know of why this work is important. This will help all readers understand and emphasize the value of your work.

Below are some editorial fixes that you should update.

Check order of citations (Weiner and Toth, 2012; Yan et al., 2014; Welch and Lister, 2014; Glastad et al., 2011)) these may need to be reordered according to journal style.

Table 1. Remove the comma after Cardoen ,
Ecdysteroid regulated gene E93/Mblk-1 transcription factor Mushroom body large-type Kenyon cell specific protein 1 -like (LOC100645656) Upregulated in honeybee reproductive workers (Cardoen,et al., 2011)

Here also

Ecdysone inducible gene L2/ ImpL2 Neural/ectodermal development factor IMP-L2-like (LOC100645498) Upregulated in honeybee nonreproductive workers (Cardoen,et al., 2011)

Page 6 Space needed between exons and (Araujo et. al)
coding exons(Araujo et al., 2012; Lytle et al., 2007)


Page 7, Figure 3. wilcoxon is a proper name, it should be Wilcoxon and the type of test. Is it signed-ranks? matched-paired signed ranks? OR Wilcoxon Mann-Whitney test?

Figure 3. Graph lines and legend should be in thicker lines and bold for ease of reading.

·

Basic reporting

The introduction is pretty sketchy in terms or providing context. Imprinting and epigenetics are said to be important for certain areas, but reasons aren’t given. Haig’s theory is mentioned but an outside reader will have no idea what that is (and, the Patten review is cited on this, doesn’t Haig deserve a citation or two for the particular ideas, not just for what imprinting is). In fact, Haig’s theory provided special motivation for studying social insects before any indications of methylation in the social Hymenoptera. There are lots of taxa that have methylation but comparatively few that have a strong theoretical expectation for imprinting. That’s what makes the social insects particularly exciting in my view.

Figure 3. Why no whiskers on the maternal plot? And can it really true that the 95% confidence interval for the Difference exactly matches the range?

I’m not very familiar with the required MIQE standards for quantitative PCR, but I don’t see any mention of these in the article.

Experimental design

The main problem with this study is mentioned by the authors in their discussion and it is a serious one. What they find is differential expression, but there are possible causes of this other than imprinting. One is that the two alleles simply differ in promoter sites, so they get expressed differently in a way that is independent of which parent they came from. The core problem of this study is that the differential expression data come from a single family. Even if the maternal allele and the paternal allele in this family have different expression, there is no way to tell if parentage is the cause. You would need many families to show this, with maternal and paternal alleles consistently showing the same direction of effect in each. Or, as some studies of imprinting do, you study reciprocal crosses to ensure that the same allele is studied both when it is inherited from the father and when it is inherited from the mother. The authors suggest that their result is still interesting because the differential expression is found in exactly the kind of gene predicted by the kin conflict theory. I see the point, but they only examined genes of that type, so any result showing differential expression (for whatever cause) would have matched a gene predicted by the kin conflict theory. If differential expression were really rare in general, then finding it in one of 12 candidate genes might be notable. But given that differential expression is quite common for a variety of reasons other than imprinting (see Palacios reference) then it would not be surprising to see an example of differential expression in a set of candidate genes, even if imprinting never occurs in this species. In the end, I don’t think the study can claim to show anything definitive about imprinting.

Validity of the findings

An important issue, about which I am not completely certain, concerns whether differential expression was actually shown, even in the single family studied. Quantitative pcr is used for this, with the key comparison being between pcr product specific to the maternal allele and one specific to the paternal allele (both subtracted from a reference product). I can accept that the measures of relative amount of product are accurate, but do they actually reflect the relative amounts of RNA template for the two alleles? I don’t see why. The forward primers are different for the two, both in their terminal base (to match the snp) and in their length. So couldn’t those amplify differently even if the templates were in the same initial concentration? An obvious control is to use the method on genomic DNA, which contains 50% of each allele, but this wasn’t done.

Comments for the author

This paper endeavors to study the possibility of genomic imprinting in a bumble bee. The kin conflict theory of the evolution of imprinting predicts that haplodiploid social insects should have rich imprinting but this idea has been little tested, so this kind of study is a great idea. However, I think this particular study is too limited to draw very useful conclusions about imprinting. At the very least, I think the study needs to add controls for the qPCR (unless I am wrong about that) and increase the sample size enough to test the effects of inheriting both snps through both parents.

Reviewer 3 ·

Basic reporting

In general, the article is well written and easy to read. The results are well presented and easy to interpret, though see my more detailed comments in the general comments to the author.

Experimental design

Please see general comments below.

Validity of the findings

Please see general comments below.

Comments for the author

This paper uses a candidate gene approach to search for evidence of genomic imprinting in bumblebees. The authors test 12 candidate genes that have previously been implicated in worker reproduction in honey bees and bumblebees. They use SSCP tests and sanger sequencing to identify polymorphisms in each candidate gene, followed by a qPCR analysis to search for allele specific expression in whole bodies of worker bumblebees. They were able to test this in 2 of 12 candidate genes, and found evidence of allele specific expression in one of these genes: ecdysone 20-monoosygenase-like. This gene is unregulated in honey bee laying workers, and combined with their evidence of allele-specific expression in Bombus, appears to be consistent with the predictions of Haig’s theory of genomic imprinting.

Overall, I find that this is an interesting study with the molecular methods clearly described and the results well articulated. However, I have a few comments about the study design that would greatly improve the quality of this work and provide convincing evidence for genomic imprinting in Bombus. That said, the authors do a decent job of discussing the limitations of their current work, though perhaps these drawbacks could do with a bit more emphasis. For example, there should be some more specific discussion on the other factors that can drive allele specific expression rather than just stating these other factors exist.

I have two main concerns:

First, this study tested for allele specific expression in a single Bombus colony. In order to find evidence for genomic imprinting, these tests need to be conducted on reciprocal crosses that can be used to uncouple allelic effects from parental effects. Only if expression is indeed dependent on parent of origin can it be concluded that there is evidence for genomic imprinting. At the very least, multiple alleles inherited from the mother or the father should be tested. Unfortunately in this study there were no additional screened colonies where a polymorphism was identified.

Second, I think the allele-specific expression results could be more clearly represented in Figure 3. What is the scale for “Difference”? What proportion of the expression is of the maternal copy vs. the paternal copy? If this is the difference in relative abundance, it’s rather hard to interpret. It would be more useful to discuss this in terms of proportions. Are we talking about a difference that looks like 90:10 maternal:paternal or more like 60:40? Imprinting is most often defined as complete silencing of one of the parental alleles. In this case, there appear to be much subtler effects on gene expression, and this should be quantified and described more clearly.

Finally, as a suggestion for future work, if the authors continue to search for evidence of genomic imprinting in this system, it might be more fruitful to look for evidence of allele-specific expression in queenless, egg-laying workers, perhaps in specific tissues (ovaries, brains) rather than in whole bodies of queenright workers.

---

## Round 0.2 · accepted · Accept

This revision address the issues identified by the reviewers.

Reviewer 1 ·

Basic reporting

Fine. Adjustments to the introduction helps this manuscript.

Experimental design

To be clear, bumble bees are my field but the research techniques used in this manuscript are not. I do not do this kind of research. That said, I feel you addressed in your rebuttal and in your revisions some of the concerns raised by the other reviewers.

Validity of the findings

I feel the ultimate value of this research lies with the readers. I also feel that, while there are limitations to what the results say based on the techniques used, you have some interesting results that can be built on through subsequent research.

Comments for the author

While I may not understand the technical aspects of the research techniques used, I do understand the implications of this work.

I also appreciate the tone of your rebuttal and your genuine attempt to take the reviewers' comments for improvement and make appropriate changes to your manuscript.

Reviewer 3 ·

Basic reporting

The revised version of this manuscript has addressed all of my previous concerns.

Experimental design

While the experimental design of the paper could have been better, the results presented are clear and concise and the discussion of these results is now much more appropriate.

Validity of the findings

No comments.

Comments for the author

No comments.